# Larvicidal Effect of *Hyptis suaveolens* (L.) Poit. Essential Oil Nanoemulsion on *Culex quinquefasciatus* (Diptera: Culicidae)

**DOI:** 10.3390/molecules27238433

**Published:** 2022-12-02

**Authors:** Taires Peniche, Jonatas L. Duarte, Ricardo M. A. Ferreira, Igor A. P. Sidônio, Rosângela S. F. R. Sarquis, Ícaro R. Sarquis, Anna E. M. F. M. Oliveira, Rodrigo A. S. Cruz, Irlon M. Ferreira, Alexandro C. Florentino, José C. T. Carvalho, Raimundo N. P. Souto, Caio P. Fernandes

**Affiliations:** 1Post-Graduate Program in Tropical Biodiversity, Federal University of Amapá, Rodovia Juscelino Kubitschek Km 02, Jardim Marco Zero, Macapá CEP 68903-419, Amapá, Brazil; 2Laboratory of Arthropoda, Collegiate of Biology, Federal University of Amapá, Rodovia Juscelino Kubitschek Km 02, Jardim Marco Zero, Macapá CEP 68903-419, Amapá, Brazil; 3Laboratory of Phytopharmaceutical Nanobiotechnology, Collegiate of Pharmacy, Federal University of Amapá, Rodovia Juscelino Kubitschek Km 02, Jardim Marco Zero, Macapá CEP 68903-419, Amapá, Brazil; 4Laboratory of Pharmaceutical Research, Collegiate of Pharmacy, Federal University of Amapá, Rodovia Juscelino Kubitschek Km 02, Jardim Marco Zero, Macapá CEP 68903-419, Amapá, Brazil; 5Laboratory of Biocatalysis and Chemical Biotransformation, Federal University of Amapá, Rodovia Juscelino Kubitschek Km 02, Jardim Marco Zero, Macapá CEP 68903-419, Amapá, Brazil; 6Graduate Program in Envirionmental Sciences (PPGCA), Laboratory of Ichthyology and Genotoxicity (LIGEN), Federal University of Amapá, Rodovia Juscelino Kubitschek Km 02, Jardim Marco Zero, Macapá CEP 68903-419, Amapá, Brazil

**Keywords:** culicidae, filariasis, hydrodistillation, miniemulsion, nanostructured system

## Abstract

Mosquitoes can be vectors of pathogens and transmit diseases to both animals and humans. Species of the genus *Culex* are part of the cycle of neglected diseases, especially *Culex quinquefasciatus*, which is an anthropophilic vector of lymphatic filariasis. Natural products can be an alternative to synthetic insecticides for vector control; however, the main issue is the poor water availability of some compounds from plant origin. In this context, nanoemulsions are kinetic stable delivery systems of great interest for lipophilic substances. The objective of this study was to investigate the larvicidal activity of the *Hyptis suaveolens* essential oil nanoemulsion on *Cx. quinquefasciatus*. The essential oil showed a predominance of monoterpenes with retention time (RT) lower than 15 min. The average size diameter of the emulsions (sorbitan monooleate/polysorbate 20) was ≤ 200 nm. The nanoemulsion showed high larvicidal activity in concentrations of 250 and 125 ppm. CL_50_ values were 102.41 (77.5253–149.14) ppm and 70.8105 (44.5282–109.811) ppm after 24 and 48 h, respectively. The mortality rate in the surfactant control was lower than 9%. Scanning micrograph images showed changes in the larvae’s integument. This study achieved an active nanoemulsion on *Cx. quinquefasciatus* through a low-energy-input technique and without using potentially toxic organic solvents. Therefore, it expands the scope of possible applications of *H. suaveolens* essential oil in the production of high-added-value nanosystems for tropical disease vector control.

## 1. Introduction

*Culex quinquefasciatus* Say 1983 is cosmopolitan, synanthropic, and well adapted to anthropized and peridomestic environments [1]. It is an important vector of filariasis in the tropics [2] and can cause alarming socio-economic impacts in the affected regions [3]. Filariasis is a neglected tropical and subtropical disease, although there has been a decrease in cases since the 2000s [4]. It is estimated to affect more than 120 million people in 72 countries [5]. In 2020, more than 800 million people in 47 countries were still infected with filariasis and needed treatment because they live in risk areas for microfilariae infection [6]. In Brazil, *Cx*. *quinquefasciatus* is a vector of lymphatic filariasis, also known as elephantiasis [7,8,9], and of the Oropouche virus (OROV), which causes Oropouche fever. This arbovirus is present mainly in the Amazon region [10]. It has also been recently found in the state of Mato Grosso [11]. Lymphatic filariasis has an exclusively urban distribution in Brazil [12,13] and has been found in the states of Pará, Amazonas, Alagoas, Bahia, and Rio Grande do Sul. Nowadays, it is limited to the Northeastern region, in the state of Pernambuco, with a large number of notified cases. The Brazilian government has created a plan to eradicate the disease, which aims at the treatment of chronic patients and the control of the vector of microfilariae [14]. The control of *Cx. quinquefasciatus* can be difficult because there might be genetically different populations of these culicides within a single country [15]. Reducing the proliferation of this species requires either biological or chemical control, in which synthetic pyrethroid, carbamate, or organophosphate insecticides are diluted and applied to breeding sites [16]. However, chemicals based on these substances can lead to the vector insect’s resistance, toxicity to non-target individuals, and the emergence of carcinogenic cells in mammals [17,18].

Natural substances originating from the secondary metabolism of plants as an environmental response are emerging as an alternative for efficient and nature-friendly natural insecticides [19]. Essential oils are products obtained from plant parts through steam distillation or from the expression of citrus fruit pericarp. In general, the chemical group of secondary metabolites that constitute essential oils is terpenoids [20], despite the fact some of them may present other classes (e.g., phenylpropanoids). These volatile substances that constitute essential oils have several biological activities regarding insects, such as toxicity, repellent, anti-oviposition, inhibition of feeding activities, and behavioral changes [21,22,23,24]. Several mechanisms might be involved. Among them are alterations in the insects’ cholinergic system [25]. Studies involving the biological effects of essential oils on disease vector insects have been developed, which report that they are potentially useful in integrated vector management [21,26].

*Hyptis suaveolens* (L.) Poit. belongs to the Lamiaceae family, which comprises several plant species with bioactive essential oils against insects [27]. *H. suaveolens* essential oil has, as its major constituents, sabinene, β-caryophyllene, 4-terpineol, and terpinolene—all terpenoids [28]. However, the composition and quantity of these constituents can vary, since the same species might have different chemotypes [29,30]. *H. suaveolens* essential oil showed larvicidal and repellent activity on insects of the Culicidae family, revealing its potential as an insecticide [31,32]. However, the intrinsically limited water solubility of constituents of essential oils impairs the development of viable larvicides containing promising essential oils.

Nanotechnology is the development of materials on a nanometer scale. It has multidisciplinary applications that contribute to technological advancement [33]. Nanoemulsions are nanostructured systems with an average droplet size diameter of the internal phase below 200 nm [34]. This system is usually stabilized with emulsifiers, reducing the interfacial tension and allowing the dispersion of one immiscible liquid into another. Nanoemulsions have kinetic stability as their main characteristic, being more resistant to the loss of stability processes such as creaming, sedimentation, and phase separation [35]. Studies show that the biological effects of essential oils are enhanced when nanoencapsulated [36]. Another advantage is that it allows lipophilic constituents to disperse in water [37], which could allow essential oils to be applied to Culicidae breeding grounds more efficiently. The bioactive properties of *H. suaveolens* on insects have been investigated in studies with extracts and essential oil, which provide incipient data on obtaining and evaluating its nanoemulsions. Therefore, in the present study, we evaluated the larvicidal activity of *H. suaveolens* essential oil nanoemulsion on *Cx. quinquefasciatus* larvae with low energy input and without solvents.

## 2. Results

### 2.1. Hyptis Suaveolens Essential Oil

The yield of essential oil obtained from *H. suaveolens* leaves was 0.31%. The chromatographic profile obtained through GC-MS analysis revealed a predominance of monoterpenes with a retention time (RT) of less than 15 min and chromatographic peaks of sesquiterpenes with retention times between 20 and 30 min (Figure 1). Table 1 shows the 26 compounds identified that constitute 99.9% of essential oil. The main components were 1,8-cineole (35.31%), fenchone (9.60%), γ-elemene (7.21%), sabinene (6.51%), limonene (5.43%), and caryophyllene (5.11%).

### 2.2. Hyptis Suaveolens Nanoemulsion

Nanoemulsions prepared with sorbitan trioleate/polysorbate 20 at HLB of 10, 11, 12, and 13 resulted in creaming after the preparation. Only at HLB of 14, 15, and 16.7 was a slightly blue reflection observed, typical of nanoemulsions due to the Tyndall effect (Figure 2). The systems prepared with sorbitan monooleate/polysorbate 20 had a thin and homogeneous aspect at HLB of 11, 12, 13, 14, and 15, evidenced by a slightly blue reflection. After seven days of storage, few changes and a low creaming formation were observed at HLB of 11-14 (Figure 3). Therefore, the nanoemulsion for the larvicidal assay had a determined rHLB of 15 (sorbitan monooleate/polysorbate 20) since it showed no signs of instability and preserved the initial characteristics after the evaluation period of storage.

#### Characterization of the Nanoemulsions

Table 2 shows the droplet size, polydispersity index, and zeta potential measurements of the formulations prepared with *H. suaveolens* essential oil and the surfactant pair of sorbitan trioleate/polysorbate 20. Emulsions prepared with these surfactants at HLB of 10, 11, 12, and 13 had an average size diameter size above 200 nm. Each size was 793.0 ± 536.9, 664.5 ± 122.2, 263.9 ± 15.36, and 255.1 ± 33.26, respectively. Thus, measurements were taken on day 0 only. In the nanoemulsions prepared with these surfactants at HLB of 14, 15, and 16.7, droplet sizes below 200 nm, a polydispersity index below 0.300, and zeta potential below −20 mV were observed during storage.

Table 3 shows the measurements for the nanoemulsions set made with the surfactant pair of sorbitan monooleate/polysorbate 20. All emulsions had measurements of ≤ 200 nm. The system prepared solely with polysorbate 20 resulted in the largest size and values were significant when compared between days 0 and 7 (*p* < 0.0001). The nanoemulsion prepared with this pair at HLB of 15 resulted in the smallest diameter (day 0 = 69.47 ± 0.6717; day 1 = 70.42 ± 0.2055; day 2 = 69.31 ± 0.1212; and day 7 = 67.79 ± 1.040) with little variation in the polydispersity index (day 0 = 0.179 ± 0.009; day 1 = 0.173 ± 0.004; day 2 = 0.177 ± 0.009; and day 7 = 0.206 ± 0.017). Moreover, there was no significant difference in size and polydispersity (*p* > 0.05), which validates the macroscopic analysis that this was the nanoemulsion with the highest stability. Size distribution and zeta potential graphs are presented in Figure 4. Therefore, the rHLB of the essential oil of *H. suaveolens* was determined as 15 and this nanoemulsion was chosen for the larvicidal assay.

### 2.3. Larvicidal Activity of the Nanoemulsion

Table 4 shows the results of the influence of the nanoemulsion of *H. suaveolens* on the mortality of *Cx. quinquefasciatus* larvae at 24 and 48 h. Low mortality was observed in the control with distilled water (2 ± 0.45%). The percentage of mortality was 100% for the 200 ppm concentration (*p* = 0.000000) in the first 24 h of the experiment. The *H. suaveolens* nanoemulsion exhibited significant larvicidal activity. Mortality rates increased alongside exposure time. The concentration of 125 ppm (*p* = 0.000000) showed a mortality rate of 78% after 48 h. The ANOVA test indicated significant differences between exposure periods (F_1_ = 36.31; *p* ≤ 0.0001) and between treatments (F_20_ = 99.35; *p* ≤ 0.0001), with R-squared values of R^2^ = 0.9583 and (adjusted) R^2^ = 0.9484. Estimated values for CL_50_ and CL_90_ (with limits below and above) were 102.41 (77.5253–149.14) ppm and 168.033 (129.738–281.681) ppm, respectively, after 24 h, and 70.8105 (44.5282–109.811) ppm and 144.947 (107.148–275.553) ppm, respectively, after 48 h. The Chi-square values were X^2^ (1) = 34.1769 (*p* = 0.0000) and X^2^ (1) = 25.4694 (*p* = 0.0000) at 24 and 48 h, respectively.

The mortality rate induced by the surfactant concentration was below 9%. The Chi-square values were X^2^ (1) = 0.137867 (*p* = 0.7104) and X^2^ (1) = 0.347238 (*p* = 0.5557) at 24 and 48 h, respectively, indicating that it is not possible to estimate the CL_50_. Furthermore, there was no significant difference between mortalities in the groups treated with surfactants under different concentrations (*p* > 0.05).

### 2.4. Morphological Study of Culex Quinquefasciatus Larvae

Electron micrograph images on *Cx. quinquefasciatus* larvae killed upon treatment with *H. suaveolens* oil nanoemulsion (Figure 5D–F) demonstrated changes in the cuticle structure, bristle shortening, cephalic capsule flattening (H), and abdomen segments’ disintegration (AB). The siphon (S) did not change noticeably. The results suggest the toxic effect of the *H. suaveolens* nanoemulsion. The control larvae remained normal in appearance (Figure 5A–C).

## 3. Discussion

### 3.1. Hyptis Suaveolens Essential Oil

The chromatographic analysis of *H. suaveolens* essential oil revealed similar main constituents to Kandpal et al. [38], who reported 1,8-cineole (17.61%) and sabinene (9.42%) among the major components. The same major compound, 1,8-cineole, was identified in three species of the genus *Hyptis*, with a higher percentage in *H. suaveolens* [39]. We observed that 1,8-cineole was the most representative substance in the *H. suaveolens* collected from the same region. However, others were either not identified or had a lower percentage in the oil composition [40]. Some chemotypes have already been described for the essential oil, such as sabinene and β-caryophyllene. However, 1,8-cineole was not found [31]. Chemical variability was found in *H. suaveolens* oil from two regions: Laos (sabinene, α-phellandrene, 1,8-cineole, β-phellandrene, and limonene) and Guinea-Bissau (sabinene, limonene, and terpinolene) [41].

### 3.2. Hyptis Suaveolens Nanoemulsion

Nanoemulsions are kinetic stable colloids constituted by two immiscible liquids often stabilized by surfactants. In the case of natural-product-based nanoemulsions, one would expect solubilization of the active secondary metabolite in inert oil or the utilization of a bioactive oil, as can be observed for essential oils [42]. The main mechanism for the destabilization of nanoemulsions is Ostwald ripening; however, an opposite thermodynamic force maintaining the droplets’ composition (Compositional ripening) has been suggested to be responsible for the enhancement of stability of essential-oil-based nanoemulsions [43]. Studies of our group showed that the alteration of droplet size distribution may also be associated with a reduction in size [44], while the utilization of a blend of surfactants from the polysorbate/sorbitan series can dramatically remedy this behavior and even induce highly stable systems with stability higher than 12 months [45]. 

The *H. suaveolens* nanoemulsion demonstrated larvicidal potential against *Cx. quinquefasciatus* in this study. Studies supporting these results have been conducted, with some species of the genus *Hyptis* being tested on the vector mosquito larvae. In studies conducted with *H. suaveolens* extracts with different solvents on *Ae. aegypti* larvae, the lowest CL_50_ and CL_90_ values for hexane were 543.66 and 3546.69 ppm, respectively, at 24 h [46]. Extracts of *H. suaveolens* (hexane, chloroform, ethyl acetate, and methanol) showed similar results to those found in this study when tested against the third larval instar of *Cx. quinquefasciatus*, with CL_50_ values of 213.09, 217.64, 167.59, and 86.93 ppm [47], respectively. The *Hyptis pectinata* and *H. fruticosa* showed CL_50_ values of 366 and 502, respectively, on *Ae. aegypti* [48]. In Pavela’s [49] studies, the synergistic effect of compounds on the mortality of *Cx*. *quinquefasciatus* larvae was observed. 1,8-cineole, limonene, and camphor were some of the aromatic evaluated compounds.

The larvicidal effect of *H. suaveolens* essential oil was observed in *Aedes albopictus*, with a mortality rate of 65% at a concentration of 250 ppm and CL_50_ of 240.3 ppm [31]. Additionally, other plant products have proven to be promising larvicides and repellents [50,51]. Nanoformulations with oil and different surfactant ratios have been studied for larvicidal activity on mosquito vectors [52]. The nanoemulsion with *Rosmarinus officinalis* essential oil containing polysorbate 20 had a mortality rate of *Ae. aegypti* larvae of 90% [53]. Results in *Ae. aegypti* (CL_50_ = 34.75) and *Cx. quinquefasciatus* (CL_50_ = 56.70 ppm) larvae were observed, with nanoemulsions of *Pterodon emarginatus* produced by the low-energy-input method [54,55]. In studies with nanomaterials, the lavender (*Lavandula officinalis*) nanoemulsion showed a mortality rate of 50% after 24 h and 90% after 24 h of exposure to *Cx. quinquefasciatus* larvae. The nanoemulsion was more effective than metal nanoparticles [56].

Nanoformulations of *Azadirachta indica* (neem)—at different proportions of oil and surfactant (polysorbate 20) (1:0.30, 1:1.5, and 1:3) and average sizes of 31.03 to 251.43 nm—were tested on *Cx. quinquefasciatus* and showed low values for CL_50_ = 11. 75 mgL^−1^ [57]. Furthermore, Sugumar and co-workers [58] observed histological changes in the gut of *Cx. quinquefasciatus* larvae that were exposed to the *Eucalyptus* nanoemulsion. However, this study employed high-energy-input methods. Changes in the gut, body, and tissue of *Cx. quinquefasciatus* larvae were observed when they were exposed to different insecticides, which suggests cell toxicity [59].

The results found in this study validate the morphological study on *Cx. quinquefasciatus* larvae, in which the larvae showed flattening of the thorax (TH) and abdomen (AB) [55]. Some similar changes were caused in the anal papillae (AP) in *Ae. aegypti* larvae that were exposed to the ethanolic extract of three species of the genus *Pipper* [60]. *Magonia pubescens* extract showed changes in the epithelial cells of the *A. aegypti* larvae’s digestive tract [61].

The nanoemulsion of *Thymus vulgaris* tested in the larvae of *Anopheles stephensi*, *Ae. aegypti*, and *Cx. tritaeniorhynchus* showed morphological changes in the cuticle of the larvae, destruction of the bristles, damage to the head, and abrupt rupture of the abdominal segments, as well as the shortening of the body [62]. Similar results were observed with *Baccharis reticularia* essential oil and d-limonene nanoemulsions. [63]. Nanostructured systems were evaluated in the *Cx. quinquefasciatus* larvae. Through scanning microscopy, changes and cuticle stiffening were observed in the larvae exposed to a lavender nanoemulsion and chitosan nanoparticles [56]. Changes in the integument can damage the larvae’s internal organs. However, it is not possible to infer that it causes mortality. There are few studies with scanning images that show the effects of plant-based products on vector mosquito larvae. The changes are commonly evidenced by histology. According to Kasai and contributors [64], penetration through the cuticle is one of the action mechanisms of insecticides. There are other factors that can be related to larvicidal activity in mosquitoes, such as enzyme inhibition [65] and deterrent effects [66,67].

## 4. Materials and Methods

### 4.1. Chemicals

The sorbitan trioleate, sorbitan monooleate, and polysorbate 20 were acquired from Praid Produtos Químicos Ltda (São Paulo, Brazil).

### 4.2. Plant material

Leaves of *Hyptis suaveolens* were collected in Macapá, AP, Brazil (00°01′33.7″ N and 51°08′56.6″ W) during the rainy season. Professor Rosângela S.F. Rodrigues Sarquis identified it. The specimen voucher was registered (18856) and filed in the herbarium of the Institute of Scientific and Technological Research of the State of Amapá (HAMAB).

### 4.3. Extraction of the Essential Oil

The essential oil of *Hyptis suaveolens* was extracted by hydrodistillating the leaves (1102.118 g) using a Clevenger-type apparatus for three hours. Then, the essential oil was collected and filtered with sodium sulfate anhydrous and stored at 4 °C.

### 4.4. Analysis of Gas-Chromatography

The essential oil of *H. suaveolens* was analyzed using a GCMS-QP5000 gas chromatograph (Shimadzu, Kyoto, Japan) equipped with a mass spectrometer. The gas chromatograph (GC) conditions were [68] an injector temperature of 260 °C, a temperature (FID) of 290 °C, helium as the carrier gas, a flow rate of 1 mL/min, and an injection rate of 1:40. The temperature started at 60 °C and increased to 290 °C at a rate of 3 °C/min. One microliter of the sample was dissolved in dichloromethane (1:100 mg/μL) and injected into a DB-5 column (0.25 mm ID, 30 m in length, 0.25 μm of film thickness). The mass spectrometer (MS) (Shimadzu, Kyoto, Japan) conditions were electron impact ionization of 70 eV and a scan rate of 1 scan/s. The retention indices (RI) were calculated by interpolating the retention times of a mixture of aliphatic hydrocarbons (C9-C30) analyzed under the same conditions [69]. The identification of substances was performed by comparing the retention indices and mass spectra with those reported in the literature [70]. The MS fragmentation pattern of the compounds was also compared with NIST mass spectrum libraries. The relative abundance of the chemical constituents was performed through flame ionization gas chromatography (GC/FID) (Shimadzu, Kyoto, Japan) under the same GC/ES conditions. Analysis and percentages of these compounds were obtained through the FID peak area normalization method.

### 4.5. Nano-Emulsions

#### 4.5.1. Assessment of the Required Hydrophile–Lipophilic Balance (rHLB) of *Hyptis suavelens* Oil

The required hydrophile–lipophile balance (HLBr) of the essential oil of *H suaveolens* was determined by mixing two pairs of non-ionic surfactants: (i) Sorbitan monooleate (HLB = 4.3)/polysorbate 20 (HLB = 16.7) and (ii) sorbitan trioleate (HLB = 1.8)/polysorbate 20. The HLBr was defined as the HLB of the surfactant mixture (10, 11, 12, 13, 14, and 15) associated with the most stable system. If polysorbate 20 allows the obtainment of the most stable system, the HLBr = HLB_polysorbate20_ = 16.7. Calculation of the HLB of a pair of surfactants can be performed as follows:(1)HLB=(HLBa×ma+HLBb×mb )/(ma+mb)

#### 4.5.2. Nano-Emulsion Method

This study used a low-energy-input method [34,71]. Each nanoemulsion consisted of 98% (*w*/*w*) water, 1% surfactant(s) (sorbitan monooleate or sorbitan trioleate/polysorbate 20), and 1% essential oil of *H. suaveolens* (final mass = 10 g). The oil phase (surfactant(s) + oil) was homogenized. Then, distilled water was added, drop by drop, under vigorous stirring using a vortex-type stirrer. Characterization was performed immediately after the preparation and after 1 and 7 days of storage at room temperature (25 ± 2 °C).

#### 4.5.3. Characterization of Hyptis Suaveolens Nano-Emulsion

The particle size distribution was assessed through the dynamic light scattering technique using a Zetasizer ZS (Malvern, UK). The nanoemulsion was diluted in deionized water (1:25 *v*/*v*) [72]. The measurements were taken in triplicate. The droplet size and polydispersity index were expressed by the mean and standard deviation.

### 4.6. Larvicidal Bioassay

*Culex quinquefasciatus* larvae were obtained from the Laboratory of Arthropoda (Federal University of Amapá, Macapá, Brazil). The biological assay was performed under controlled conditions, having kept the larvae at 25 ± 2 °C, a relative humidity of 75 ± 5%, and a 12 h light/dark cycle. The experimental evaluation was performed according to the World Health Organization (WHO) [73] protocol with some modifications. The experimental design was completely randomized in five replicates, with 10 fourth-instar larvae in each sample. The nanoemulsions were diluted in distilled water at 15.625, 31.25, 62.25, 125, and 250 ppm (concentrations refer to *H. suaveolens* essential oil in an aqueous medium). Solutions of the surfactants used in the evaluated nanoemulsion were diluted in distilled water at the same concentrations for control. The negative control group consisted of distilled water. Mortality rates were registered after 24 and 48 h of exposure.

### 4.7. Morphological Study of Culex Quinquefasciatus Larvae

Larvae of the treatment and control group were fixed in formaldehyde (10%), and the external morphology was checked under a low vacuum using a scanning electron microscope (Tabletop Microscope TM3030Plus, Hitachi, Tokyo, Japan).

### 4.8. Statistical Analysis

Before the analyses, the mortality rate in the treatments was corrected to that of the controls using Abbott’s formula [74]. Mortality data were subjected to Probit analysis to estimate CL_50_ and CL_90_ values using the software Statgraphics Plus V 5.1 (Stat Easy Co., Minneapolis, MN, USA). A two-way ANOVA test and a subsequent Tukey’s HSD test were performed using the software R [75], and differences were considered significant when *p* ≤ 0.05.

## 5. Conclusions

Natural products can be considered alternatives to synthetic insecticides for vector control. Due to the low solubility in water of many substances from the secondary metabolism of plants, such as the constituents of essential oils, the preparation of oil-in-water nanoemulsions is promising. This study achieved an optimal nanoemulsion through the utilization of two blends of non-ionic surfactants that led to the determination of the required hydrophile–lipophile balance of the *H. suaveolens* essential oil. The active nanoemulsion against *Culex quinquefasciatus*, which presented a narrow size distribution after its preparation by a low-energy-input technique and without the use of potentially toxic organic solvents, expands the scope of possible applications of *Hyptis suaveolens* essential oil in the production of colloidal nanosystems with high added value for tropical disease vector control, contributing to the state of the art of phytopharmaceutical nanobiotechnology.

## Figures and Tables

**Figure 1 molecules-27-08433-f001:**
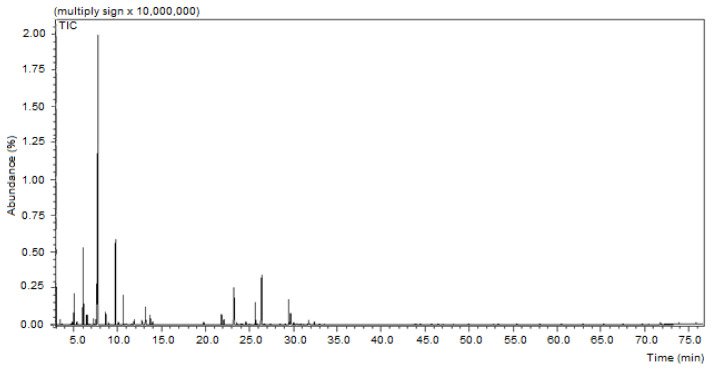
Chromatogram of *Hyptis suaveolens* essential oil obtained through gas chromatography and mass spectrometry analysis.

**Figure 2 molecules-27-08433-f002:**
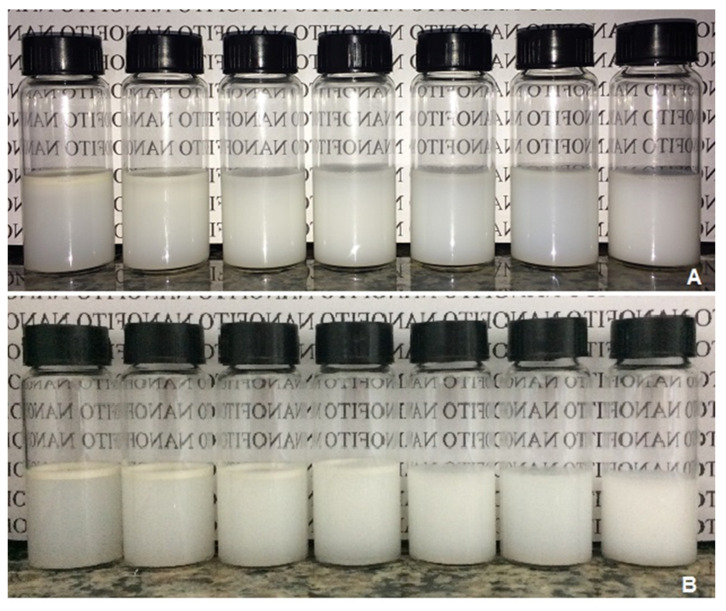
Set of emulsions prepared with *H. suaveolens* essential oil and the surfactants sorbitan trioleate/polysorbate 20 at different rates (from right to left: HLB of 10, 11, 12, 13, 14, 15, and 16.7). (**A**) Day 0 of preparation. (**B**) After 7 days of storage.

**Figure 3 molecules-27-08433-f003:**
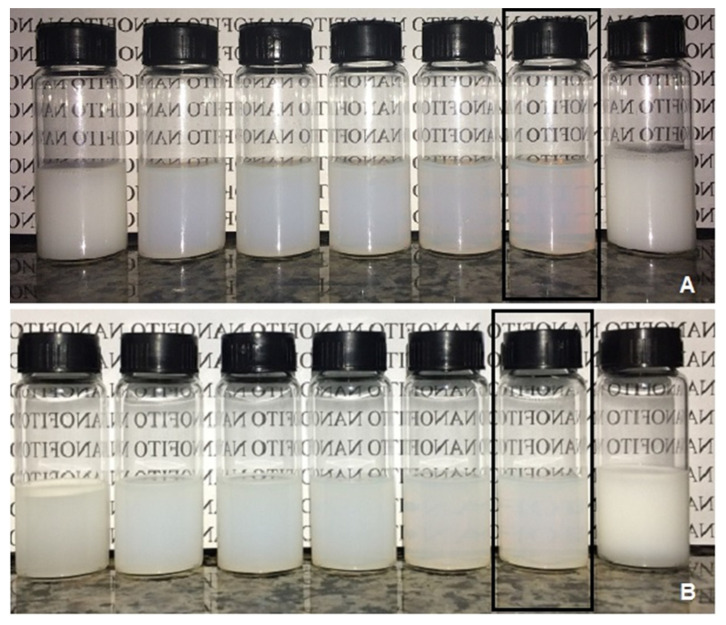
Set of emulsions prepared with *H. suaveolens* essential oil and the surfactants sorbitan monooleate/polysorbate 20 at different rates (from right to left: HLB of 10, 11, 12, 13, 14, 15, and 16.7). Highlighted is the rHLB of 15 since this is the HLB value of the mixture of surfactants that allowed the most stable nanoemulsion, with this being chosen for the bioassay. (**A**) Day 0 of preparation. (**B**) After 7 days of storage.

**Figure 4 molecules-27-08433-f004:**
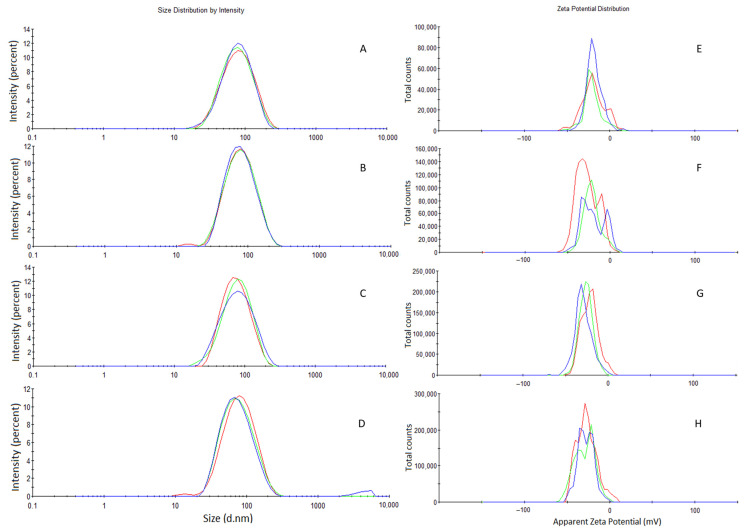
Size distribution graphs: (**A**) Day 0, (**B**) day 1, (**C**) day 2, and (**D**) day 7. Zeta potential (**E**) day 0, (**F**) day 1, (**G**) day 2, and (**H**) day 7 of the nanoemulsion prepared with *H. suaveolens* (rHLB = 15) and mixture of sorbitan monooleate/polysorbate 20 at HLB of 15.

**Figure 5 molecules-27-08433-f005:**
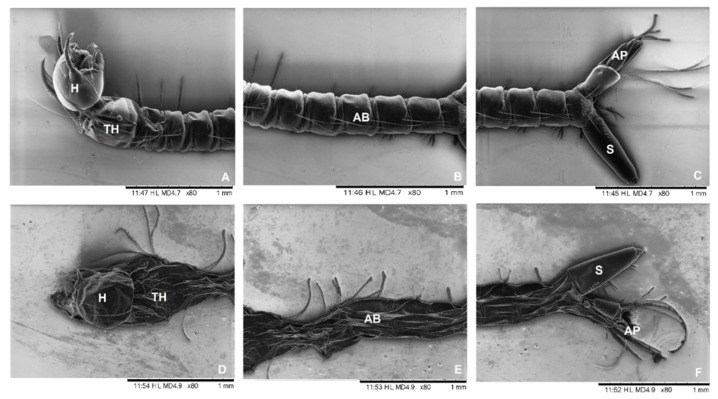
Scanning electron micrographs of *Culex quinquefasciatus* larvae. Normal-looking larvae in the control group (**A**–**C**). Changes shown in larvae treated with *H. suaveolens* nanoemulsion (250 ppm) (**D**–**F**). Damage to the entire length of the cuticle, except for the siphon and the head. Head (H), thorax (TH), abdomen (AB), siphon (S), and anal papillae (AP).

**Table 1 molecules-27-08433-t001:** Chemical constituents of *Hyptis suaveolens* essential oil.

Peak	RT (min)	Compounds	(%) GC-MS	RI Exp. *	RI Lit. **
1	3.459	(*Z*)-3-hexen-1-ol	0.22	846	857
2	5.031	δ-Pinene	2.26	930	939
3	5.402	Camphene	0.18	949	953
4	6.026	Sabinene	6.51	974	976
5	6.131	β-Pinene	4.96	979	980
6	6.463	β-Myrcene	0.87	992	990
7	6.577	Octan-3-ol	0.82	997	993
8	7.260	α-Terpinene	0.55	1018	1018
9	7.515	*o*-Cymene	0.48	1025	1022
10	7.664	Limonene	5.43	1030	1029
11	7.780	1,8-cineole	35.31	1033	1033
12	8.649	γ-Terpinene	1.15	1059	1062
13	9.728	Fenchone	9.60	1091	1086
14	10.658	Fenchol	3.17	1118	1117
15	11.867	Camphor	0.59	1146	1143
16	12.733	Borneol	0.72	1168	1165
17	13.179	4-Terpineol	2.01	1179	1177
18	13.730	α-Terpineol	1.22	1193	1134
19	21.794	Endo-Bourbonanol	1.47	1386	1515
20	22.105	β-Elemene	0.68	1393	1339
21	23.220	Caryophyllene	5.11	1421	1466
22	25.706	Germagrene	3.05	1483	1503
23	26.359	γ-Elemene	7.21	1499	1433
24	29.494	Spathulenol	3.72	1582	1578
25	29.693	Caryophyllene oxide	2.10	1586	1583
26	31.780	Elemol	0.60	1641	1549
TOTAL			99.9%		

* RI exp: Calculated RI. ** RI lit: Tabulated RI for the compound.

**Table 2 molecules-27-08433-t002:** Average diameter size, polydispersity index, and zeta potential (mV) of emulsions with *H. suaveolens* essential oil and the surfactant pair sorbitan trioleate/polysorbate 20 (HLB of 10, 11, 12, 13, 14, 15, and 16.7).

	DAY 0			DAY 1			DAY 2			DAY 7		
	Size (nm)	PDI	Zeta Potential (mV)	Size (nm)	PDI	Zeta Potential (mV)	Size (nm)	PDI	Zeta Potential (mV)	Size (nm)	PDI	Zeta Potential (mV)
HLB 10	793.0 ± 536.9	0.961 ± 0.068	−37.6 ± 0.814	-	-	-	-	-	-	-	-	-
HLB 11	664.5 ± 122.2	0.866 ± 0.167	−33.9 ± 0.306	-	-	-	-	-	-	-	-	-
HLB 12	263.9 ± 15.36	0.638 ± 0.070	−31.0 ± 1.01	-	-	-	-	-	-	-	-	-
HLB 13	255.1 ± 33.26	0.472 ± 0.052	−32.7 ± 0.458	-	-	-	-	-	-	-	-	-
HLB 14	144.0 ± 1.358	0.258 ± 0.003	−32.2 ± 0.252	143.0 ± 0.5859	0.251 ± 0.002	−28.9 ± 1.31	140.5 ± 0.2517	0.249 ± 0.017	−30.7 ± 0.404	142.9 ± 1.967	0.256 ± 0.010	−36.2 ± 0.04899
HLB 15	114.2 ± 1.069	0.222 ± 0.008	−23.8 ± 0.757	114.0 ± 1.206	0.221 ± 0.010	−19.5 ± 0.346	115.7 ± 0.05774	0.212 ± 0.008	−25.9 ± 1.10	112.2 ± 0.1528	0.220 ± 0.007	−27.8 ± 0.889
HLB 16.7	146.3 ± 0.6506	0.144 ± 0.012	−23.9 ± 1.95	160.8 ± 1.229	0.161 ± 0.013	−21.2 ± 1.81	169.3 ± 0.8660	0.164 ± 0.006	−21.5 ± 0.436	146.7 ± 0.05774	0.149 ± 0.007	−21.8 ± 0.850

Polydispersity index (PDI). Data are expressed as the mean and standard deviation (n = 3, mean ± SD).

**Table 3 molecules-27-08433-t003:** Average diameter size, polydispersity index, and zeta potential (mV) of emulsions with *H. suaveolens* essential oil and the surfactant pair sorbitan monooleate/polysorbate 20 (HLB of 10, 11, 12, 13, 14, 15, and 16.7).

	DAY 0			DAY 1			DAY 2			DAY 7		
	Size (nm)	PDI	Zeta Potential (mV)	Size (nm)	PDI	Zeta Potential (mV)	Size (nm)	PDI	Zeta Potential (mV)	Size (nm)	PDI	Zeta Potential (mV)
HLB 10	184.7 ± 8.697	0.402 ± 0.005	−50.5 ± 1.08	185.6 ± 4.046	0.441 ± 0.043	−50.5 ± 0.656	198.7 ± 6.200	0.433 ± 0.074	−47.5 ± 1.36	171.3 ± 4.932	0.442 ± 0.021	−43.3 ± 0.624
HLB 11	75.97 ± 0.2203	0.130 ± 0.005	−37.3 ± 2.91	76.30 ± 0.5577	0.142 ± 0.011	−48.1 ± 3.50	76.59 ± 0.4508	0.145 ± 0.015	−45.8 ± 2.44	77.43 ± 0.09539	0.144 ± 0.011	−41.0 ± 2.41
HLB 12	77.70 ± 0.2570	0.194 ± 0.013	−34.8 ± 1.46	78.04 ± 0.6915	0.198 ± 0.002	−43.6 ± 4.16	79.14 ± 0.7842	0.207 ± 0.011	−40.7 ± 1.55	78.26 ± 0.2750	0.188 ± 0.003	−45.4 ± 0.985
HLB 13	81.95 ± 0.2793	0.166 ± 0.009	−19.4 ± 0.764	82.73 ± 0.2581	0.169 ± 0.007	−23.8 ± 1.42	81.78 ± 0.5977	0.175 ± 0.004	−26.5 ± 0.702	83.02 ± 0.8981	0.172 ± 0.005	−29.9 ± 1.36
HLB 14	78.14 ± 0.2346	0.244 ± 0.006	−25.0 ± 1.06	76.87 ± 0.4140	0.232 ± 0.006	−30.1 ± 3.38	76.29 ± 0.1970	0.230 ± 0.005	−29.2 ± 4.08	74.96 ± 0.3057	0.236 ± 0.008	−34.4 ± 3.52
HLB 15	69.47 ± 0.6717	0.179 ± 0.009	−19.4 ± 1.14	70.42 ± 0.2055	0.173 ± 0.004	−22.7 ± 3.78	69.31 ± 0.1212	0.177 ± 0.009	−25.6 ± 3.77	67.79 ± 1.040	0.206 ± 0.017	−28.4 ± 1.06
HLB 16.7	134.4 ± 1.300	0.163 ± 0.017	−25.2 ± 1.23	173.4 ± 1.436	0.191 ± 0.015	−28.7 ± 1.27	173.5 ± 1.234	0.175 ± 0.019	−23.7 ± 0.153	162.6 ± 0.8505	0.182 ± 0.013	−24.4 ± 0.306

Polydispersity index (PDI). Data are expressed as the mean and standard deviation (n = 3, mean ± SD).

**Table 4 molecules-27-08433-t004:** Mortality rate (%) of *Culex quinquefasciatus* larvae after treatment with the nano-emulsion of *Hyptis suaveolens* essential oil (rHLB = 15).

Exposure Time (h)	Control (Distilled Water)	CONCENTRATIONS				
		15.625 ppm	31.25 ppm	62.5 ppm	125 ppm	250 ppm
24	0 ^a^	2 ± 0.45 ^a^	6 ± 0.55 ^b^	32 ± 0.84 ^c^	62 ± 0.84 ^d^	100 ± 0 ^e^
48	2 ± 0.45 ^a^	12 ± 0.45 ^b^	20 ± 0.71 ^b^	62 ± 0.84 ^c^	78 ± 0.84 ^cd^	100 ± 0 ^de^

Values are the mean of the five replicates and standard deviation (mean ± SD). Mean in the same row with different superscripts indicate significant difference (*p* < 0.05).

## Data Availability

Not applicable.

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
