# Peer review of "Larvicidal Effect of Hyptis suaveolens (L.) Poit. Essential Oil Nanoemulsion on Culex quinquefasciatus (Diptera: Culicidae)"

_molecules, 2022, doi:10.3390/molecules27238433_

Round 1
Reviewer 1 Report
1. Line 27, In abstract, Cx. quinquefasciatus, give this abbreviation before in the text.
2. Line 323, HLB of 10, 11, 12, 13, 14, 15, and 16.7 were selected for the development of nanoemuslsion, how authors calculated HLB value, elaborate with formula in the manuscript. Also, some places authors mentioned HLBr and some places HLB, write uniformly in the text.
3. Authors used essential oil for the preparation of nanoemulsion alone, essential oil may not form good nanoemulsion with a fixed oil, as figure 3 is clearly showing instability at 7th day of storage. Why did not authors use fixed oil to stabilize the nanoemulsion?
4. Why did not authors perform TEM analysis of nanoemulsion to check the globules formation, morphology, and shape, also FTIR study is required to confirm the loading of essential?
5. Authors selected polysorbate 20 as a surfactant which is a non-ionic surfactant, what could be the reasons for negative value of zeta potential?
6. Provide the particle size and zeta potential analysis distribution curve of first and seven days results.
Author Response
- Line 27, In abstract, Cx. quinquefasciatus, give this abbreviation before in the text.
Answer: We appreciate the correction, which was performed in the text.
- Line 323, HLB of 10, 11, 12, 13, 14, 15, and 16.7 were selected for the development of nanoemulsion, how authors calculated HLB value, elaborate with formula in the manuscript. Also, some places authors mentioned HLBr and some places HLB, write uniformly in the text.
Answer: We appreciate the comment and included the formulae in the manuscript. Text was also modified for better understanding of Molecules readers. We would like to clarify that HLB is an empiric value attributed to surfactants. However, based on the preparation of set of emulsions with pair of surfactants, considering the generation of a more stable system, it can be suggested a required HLB (rHLB) value of the oil, defined as the HLB of surfactant (s) capable of generation this stable system. We really thank for the observation and expect that now is more clear for the readers.
- Authors used essential oil for the preparation of nanoemulsion alone, essential oil may not form good nanoemulsion with a fixed oil, as figure 3 is clearly showing instability at 7th day of storage. Why did not authors use fixed oil to stabilize the nanoemulsion?
Answer: We appreciate this relevant comment and would like to clarify. The nanoemulsion prepared with monooleate/polysorbate 20 at HLB of 15, which was defined as the rHLB of the essential oil of H. suaveolens was considered stable in the macroscopical evaluation (Fig 3). The different light for the photos may suggest difference in the macroscopical aspect, but in fact, they mantained the appearance (slight turbid with bluish reflect) in accordance with this type of colloid. Moreover, the maitainance of size distribution, as evidenced by table 3 confirm this result. We opted to use the formulation constituted by 1% of essential oil and surfactant to oil ratio of 1:1, since the size below 100 nm was suitable for the text and based in our experience, some essential oil-based nanoemulsions at higher proportions of surfactant may facilitate migration of compound from the internal phase, therefore changing size. Despite this study was properly designed and achievement of results related to development of a system by changing surfactants and choosing one for the biological assay can be observed, the author comment is very relevant and a novel investigation changing proportions of essential oil and surfactants (eg. by using pseudoternary diagram) can be performed and we expect that this will further be performed, generating other work or great interest.
- Why did not authors perform TEM analysis of nanoemulsion to check the globules formation, morphology, and shape, also FTIR study is required to confirm the loading of essential?
Answer: We appreciate the comment and would like to justify the utilization of DLS and zeta potential analysis. These are techniques widely spread for colloids and of great interest for comparing critical parameters of nanoemulsions. TEM is widely used for morphology of systems, more specifically, Cryo-TEM for nanoemulsions. However, the aim of this study was to develop the larvicidal nanoemulsion and investigate their properties for the aforementioned techniques. We expect that further study, with the H. suaveolens nanoemulsion aiming deeper investigation of larvicidal activity, for example release of compounds and residual activity, may include the morphology of the nanostructures, including in the assay media, using the Cryo-TEM technique that provide suitable conditions for not destroyng the structures and therefore analysing the nanodroplets.
- Authors selected polysorbate 20 as a surfactant which is a non-ionic surfactant, what could be the reasons for negative value of zeta potential?
Answer: We appreciate this very interesting comment. Despite the surfactants used for the larvicidal nanoemulsion (polysorbate 20/sorbitan monooleate at a blend with HLB of 15) are non ionic molecules, studies of herbal nanoemulsions reveal the ions can be absorbed to the interface and therefore modify zeta potential. This was related to conjugated bases of terpenes and or hydroxyl groups, therefore considering the complexicity of an essential oil, this can be a possible explanation for the phenomena.
- Provide the particle size and zeta potential analysis distribution curve of first and seven days results.
Answer: We appreciate the comments and would like to indicate that the graphs were included in the manuscript.
Reviewer 2 Report
The manuscript submitted for publication deals with the development and characterization of Hyptis suaveolens essential oil nanoemulsions and the evaluation of their larvicidal activity on Cx. quinquefasciatus.
The selected topic is pertinent. The structure and the content of the manuscript are convincing, with consistent and robust results. Furthermore, in general, the paper is fairly well-written.
I suggest that this manuscript should be accepted for publication after amending the suggested corrections.
Please consider the following points:
- Line 4 – Check the numbers of authors’ affiliations;
- Line 105 – Express the extraction yield of the EO in %;
- Line 221 – Add the “surname of first author et al.” before reference 38;
- Line 274 – Clarify the following sentence “The same results were identified in Ae. aegypti larvae exposed to nanoemulsion and limonene from Baccharis reticularia”. The observed results were with Baccharis reticularia essential oil and d-limonene nanoemulsions.
- In the Discussion section (3.2.), it must be discussed the stability of the nanoemulsions, as it seems that long-term stability assays haven´t been carried out;
- In the Conclusions section, the main results achieved must be pointed out.
With thanks and best wishes,
The Reviewer
Author Response
The manuscript submitted for publication deals with the development and characterization of Hyptis suaveolens essential oil nanoemulsions and the evaluation of their larvicidal activity on Cx. quinquefasciatus.
The selected topic is pertinent. The structure and the content of the manuscript are convincing, with consistent and robust results. Furthermore, in general, the paper is fairly well-written.
I suggest that this manuscript should be accepted for publication after amending the suggested corrections.
Please consider the following points:
- Line 4 – Check the numbers of authors’ affiliations;
Answer: We appreciate all the kindly comments of the referee and regarding this specific point, we corrected the authors’ affiliations.
- Line 105 – Express the extraction yield of the EO in %;
Answer: We appreciate the correction, which was performed in the text.
- Line 221 – Add the “surname of first author et al.” before reference 38;
Answer: We appreciate the correction, which was performed in the text.
- Line 274 – Clarify the following sentence “The same results were identified in Ae. aegypti larvae exposed to nanoemulsion and limonene from Baccharis reticularia”. The observed results were with Baccharis reticularia essential oil and d-limonene nanoemulsions.
Answer: We appreciate the correction, which was performed in the text.
- In the Discussion section (3.2.), it must be discussed the stability of the nanoemulsions, as it seems that long-term stability assays haven´t been carried out.
Answer: We appreciate the comment and included a more suitable discussion about stability of nanoemulsions. Please, do not hesitate to contact us if any other reference is need.
- In the Conclusions section, the main results achieved must be pointed out.
Answer: We appreciate the comment and performed a revision of the conclusion section.